

# Reliability and validity of the Körperkoordinationstest Für Kinder in Chinese children

Kai Li[1,*], Ran Bao[2,3,*], Hyunshik Kim[4], Jiameng Ma[4], Ci Song[5], Sitong Chen[6] and Yujun Cai[1]

[1] School of Physical Education, Shanghai University of Sport, Shanghai, China
[2] College of Human and Social Futures, University of Newcastle, New South Wales, Australia
[3] Active Living Research Program, Hunter Medical Research Institute, New South Wales, Australia
[4] Faculty of Sports Education, Sendai University, Shibata-machi, Japan
[5] Shanghai Datong High School, Shanghai, China
[6] Institute for Health and Sport, Victoria University, Melbourne, Australia
* These authors contributed equally to this work.

Corresponding author
Yujun Cai, caiyujun@sus.edu.cn

## ABSTRACT

**Background:** The Körperkoordinationstest Für Kinder (KTK) is a reliable and low-cost motor coordination test tool that has been used in several countries. However, whether the KTK is a reliable and valid instrument for use in Chinese children has not been assessed. Additionally, because the KTK was designed to incorporate locomotor, object control, and stability skills, and there is a lack of measurement tools that include stability skills assessment for Chinese children, the KTK's value and validity are worth discussing.

**Methods:** A total of 249 primary school children (131 boys; 118 girls) aged 9–10 years from Shanghai were recruited in this study. Against the Test of Gross Motor Development-3 (TGMD-3), the concurrent validity of the KTK was assessed. We also tested the retest reliability and internal consistency of the KTK.

**Results:** The test–retest reliability of the KTK was excellent (overall: r = 0.951; balancing backwards: r = 0.869; hopping for height: r = 0.918; jumping sideways: r = 0.877; moving sideways: r = 0.647). Except for the boys, the internal consistency of the KTK was higher than the acceptable level of Cronbach's α > 0.60 (overall: α = 0.618; boys: α = 0.583; girls: α = 0.664). Acceptable concurrent validity was found between the total scores for the KTK and TGMD-3 (overall: r = 0.420, *p* < 0.001; boys: r = 0.411, *p* < 0.001; girls: r = 0.437, *p* < 0.001).

**Discussion:** The KTK is a reliable instrument for assessing the motor coordination of children in China. As such, the KTK can be used to monitor the level of motor coordination in Chinese children.

## INTRODUCTION

Motor competence refers to motor performance, proficiency, ability, or coordination, and is defined as proficiency in performing a wide array of goal-directed motor skills as well as

the underlying mechanisms (*e.g.*, quality of movement, coordination, and control) (*Barnett et al., 2022*; *Burton & Rodgerson, 2001*; *Coppens et al., 2021*; *D'Hondt et al., 2013*). Motor competence's importance lies in its beneficial impacts on physical and mental health in children and adolescents, including behavioural (*e.g.*, promoting physical activity and reducing sedentary behaviour (*Tsuda et al., 2020*)), physiological (*e.g.*, improving physical fitness (*Utesch et al., 2019*) and improving weight status (*D'Hondt et al., 2013*)), cognitive (*e.g.*, improving cognitive function (*Haapala, 2013*)), and psychological benefits (*e.g.*, promoting perceived motor competence; *Lubans et al., 2010*). Motor coordination, an important component of motor competence, is closely associated with numerous health outcomes, while poor motor coordination has detrimental effects on overall functioning, emotional and social development (*De Chaves et al., 2016*), as well as physical activity and physical fitness in children (*Rivilis et al., 2011*). The development of motor coordination is dependent on neuromuscular and biological maturation (*De Chaves et al., 2016*). A coordinated movement pattern is the foundation of an effective execution of motor skills and is at the core of fundamental motor skills, such as locomotor skills, object control skills, and stability skills (*Coppens et al., 2021*; *Novak et al., 2016*).

Motor coordination is not a single physical fitness skill, but the synthesis of balance, rhythm, strength, lateral, speed, agility, and other human movement abilities (*Coppens et al., 2021*). It is hard to evaluate motor coordination independently from other pure fitness characteristics such as flexibility, speed, and strength (*Vandorpe et al., 2010*). A reliable and valid tool specifically designed for motor coordination would be useful for screening motor competence in children, especially in a school setting (*Vandorpe et al., 2010*). The Test of Gross Motor Development-3 (TGMD-3) (*Ulrich, 2020*) is a process-oriented assessment that focuses on assessing the quality of motor competence (*e.g.*, mechanics of movement), but it is more suitable for intervention studies because analysing video data is time consuming and costly (*Bardid et al., 2019*). Product-oriented measures focus on the outcomes of motor competence (*e.g.*, the number of tosses, or the distance of throwing), which require limited resources (*Bardid et al., 2019*). As a type of product-oriented measure, the Körperkoordinationstest Für Kinder (KTK) can assess motor coordination for both typically developing children and special children (*Vandorpe et al., 2010*).

To date, multiple motor coordination test batteries have been developed for assessment and monitoring at different life stages. The KTK is one of the most popular measurement tools, especially for children and adolescents (*Cattuzzo et al., 2017*; *Moreira et al., 2019*). The KTK was initially developed to assess global motor coordination and physical fitness (*e.g.*, body coordination) (*Rudd et al., 2016*) and was modified and used for screening motor competence in sports such as soccer (*Deprez et al., 2015*; *Vandendriessche et al., 2012*), volleyball (*Pion et al., 2015*), and figure skating (*Mostaert et al., 2016*). The KTK is comprised of four items: balancing backwards (BB), moving sideways (MS), hopping for height (HH), and jumping sideways (JS). All age groups (5–15 years) use the same items to assess motor coordination, which makes the tool suitable for longitudinal studies in samples of children and adolescents. The KTK is a simple and time-efficient assessment; children can complete all measures within 15 min. One of the main differences between
the KTK and other measurement tools is that it provides an objective and direct assessment of motor coordination (*Vandorpe et al., 2010*). In addition, the KTK assesses locomotor skills and object control skills, which are the focus of current motor competence assessment tools such as the TGMD-3, as well as stability skills (*e.g.*, balance, twisting). Unfortunately, however, few studies have focused on the measurement of stability skills in Chinese children.

The KTK has undergone reliability and validity tests in several countries, including Germany (*Kiphard & Schilling, 2007*), Brazil (*Draghi, Cavalcante Neto & Tudella, 2020*; *Moreira et al., 2019*), and Belgium (*Coppens et al., 2021*), and is used extensively to assess the motor coordination of typically-developing children (*Iivonen, Kaarina Sääkslahti & Laukkanen, 2015*; *Vandorpe et al., 2010*). Although the psychometric structure of the KTK was found to be similar across studies, different raw scores were reported in different countries (*Bardid et al., 2015*; *Liu, Chen & Cai, 2022*). For example, evidence showed that Chinese children (9–10 years) had a lower level of motor coordination than Australian and Belgian children (6–8 years) (*Liu, Chen & Cai, 2022*). Notably, there was no analysis performed to determine whether the KTK's lack of reliability and validity in Chinese children contributed to the difference in the raw scores. Scientific evidence has indicated that the differences in raw scores can be explained by the variety of physical activity contexts (*i.e.*, physical education) that children receive in primary school, which may have influenced the performance of novel motor tasks among children (*Bardid et al., 2015*). Therefore, it is essential to verify the suitability of measurement tools when they are used across different cultural backgrounds (*Cicchetti & Rourke, 2004*; *Vallerand, 1989*). If an assessment is unreliable, it will inevitably produce the wrong results, potentially leading to misdiagnosis, false alarms, or failure to detect a disorder, thus losing its value and meaning (*Valentini, Ramalho & Oliveira, 2013*).

Additionally, the other focus of this study was to determine whether gender differences across different cultural contexts affected the reliability and validity of the KTK, as research has shown that motor coordination can differ between boys and girls of the same age (*Lopes et al., 2011*; *Olesen et al., 2014*; *Re et al., 2018*; *Vandorpe et al., 2010*). Multiple differences between boys and girls, both in physical development and motor development, have been recently reported (*Goodway & Gallahue, 2020*; *Haywood & Getchell, 2020*). This could help explain the gender differences of motor coordination in children, which may also influence the suitability of the KTK items on different genders when used in a Chinese context. Unfortunately, very few studies have considered these potential gender differences when the KTK is used in a different cultural contexts.

Using reliable and valid assessments can help facilitate cross-cultural comparisons and provide a better understanding of the global level of motor coordination (*Bhui et al., 2003*). Given the lack of measurement tools used to assess the stability skills in Chinese children, the aims of this study were to: (1) investigate the level of motor coordination measured by the KTK in Chinese children; (2) examine the reliability and validity of the KTK in Chinese children; and (3) test whether gender differences in Chinese children affected the reliability and validity of the KTK.

## MATERIALS AND METHODS

### Participants

The participants for this study were recruited using a convenience sampling method. G*Power 3.0 software (University of Düsseldorf, Düsseldorf, Germany) was used to calculate the sample size (effect size = 0.25, α = 0.05, 1−β = 0.95 and a 2-tailed correlation), and the required sample size was 197. The children included in this study did not have physical and intellectual disabilities.

A total of 283 school-aged children between 9 and 10 years of age from one primary school in Shanghai, China, were invited to participate in this study. A total of 249 participants (131 boys and 118 girls) completed the study assessment and provided valid data. Before the assessments, all children were required to assent and their parents were required to provide informed consent. This study was approved by the Institutional Review Board (IRB) of the Shanghai University of Sport (102772021RT072).

### Instruments and assessments

Motor coordination was assessed using the KTK developed by *Kiphard & Schilling (1974, 2007)*. The KTK consists of four main components and the test flow is shown in Fig. 1.

The first test was BB, which evaluates balance control and coordination in the progressive recognition of the support base. Participants stepped back three times on three balance beams of different widths, each 3 m long and 8 cm high, with widths decreasing as the test progressed (6.0, 4.5, and 3.0 cm, respectively). A maximum of eight steps could be taken for each beam in each test, and a maximum of 72 steps (eight steps * three times * three beams) could be taken for the total test score. The test score was the sum of the number of test steps.

The second test was HH, which evaluates lower limb coordination, strength, and dynamic stability control. After a short run-up (about 1.5 m), participants jumped with one leg over a growing pile of pillows (60 cm * 20 cm * 5 cm each). During the whole test, the other leg could not touch the ground. *Kiphard & Schilling (1974)* set the initial height of the jump pillows (6 years: 5 cm/one piece, 7–8 years old: 15 cm/three pieces, 9–10 years old: 25 cm/five pieces. 11–14 years old: 35 cm/seven pieces) according to the age of the participants. Participants who performed successfully on the first, second, or third trials were awarded three, two, or one point(s), respectively. If participants did not succeed in the initial height test, the height was lowered by 5 cm until they succeeded. Each successful jump was followed by adding a pillow, and the test ended when participants failed three times. A maximum of 39 points (ground level + 12 pillows) could be scored for each leg, with a possible maximum score of 78 points. The test score was the sum of the points achieved by the left and right feet.

The third test was JS, which evaluates the bilateral symmetrical motor coordination, speed, and dynamic balance of the lower limbs. Participants jumped over a square wooden slat (60 cm * 4 cm * 2 cm) with both feet horizontally from left and right as much as possible within 15 s, two times. The test score added the number of jumps between the two tries.
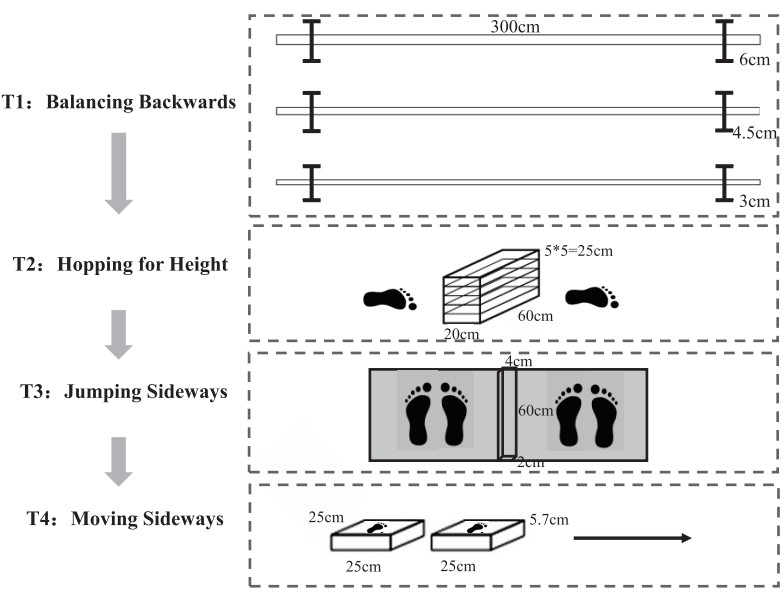

**Figure 1 The test protocol of the KTK.**

The fourth and final test was MS, which evaluates the coordination and agility of lateral movement. The test combines the velocity of the upper and lower limbs with fluidity of movement, laterality, and spatiotemporal structure. Participants stood on two side-by-side platforms (25 cm * 25 cm * 5.7 cm) and moved the two platforms by hands as fast as possible within 20 s. Each participant was given two tries, one for each of the left and right directions. The test score was the sum of the two trials.

The sum of the scores was calculated from the sum of the raw scores for the subtests (TS = BB + HH + JS + MS). All assessments followed the guidelines established by the researchers (*Kiphard & Schilling, 2007*).

Children in this study were also assessed for motor competence using the TGMD-3 (*Ulrich, 2020*). The TGMD-3 is a widely used assessment for children aged 3–10 years (*Bolger et al., 2021*; *Maïano et al., 2021*; *Webster & Ulrich, 2017*). Specifically, the TGMD-3 consists of six locomotor skills (run, gallop, hop, skip, horizontal jump, and slide) and seven ball skills (two-hand strike of a stationary ball, one-hand forehand strike of self-bounced ball, one-hand stationary dribble, two-hand catch, kick a stationary ball, overhand throw, and underhand throw) (*Ulrich, 2020*; *Webster & Ulrich, 2017*). The TGMD-3 is a reliable (ICC = 0.741–0.755) and valid ($X^2_{(64)}$ = 103, $p < 0.01$) instrument used to assess fundamental movement skills in Chinese children (*Xingying et al., 2022*; *Zhang & Cheung, 2019*).

Each test of the TGMD-3 consists of three to five performance criteria. In general, these performance criteria represent mature patterns of skills. Each performance criterion is scored as: 1 = performs correctly and 0 = did not perform correctly. The performance criteria score was calculated by adding the scores for each performance criterion in Trials 1 and 2. Skill scores were calculated by summing all performance standard scores for each skill. All assessments in this research followed the guidelines established by the authors

(*Kiphard & Schilling, 2007*; *Ulrich, 2020*). Two independent raters completed the assessment and the inter-rater reliability between them for the 13 skills ranged from 0.699 to 0.747.

## Procedure

A test–retest design was used to assess the KTK across a two-week interval, and 69 children were included in the test–retest reliability design. On both occasions, children's motor competence was assessed in the same contexts, including silent demonstrations, assessment time, and facilities.

After the test–retest was completed, the formal test was carried out. A total of 249 children completed all the formal tests. For the formal test procedure, a group of students completed the KTK test and then took the TGMD-3. All tests were conducted based on the author's operating manual (*Ulrich, 2020*).

Before the test, a five-hour workshop was delivered to the research assistants (RAs), and an assessment manual was also distributed to each RA. All tests were conducted on a sports court, and nine children in each group were assessed by two to three RAs. At the beginning of each test, one trained RA provided a silent demonstration of the skill to be tested for participants. All children performed a familiarisation trial of each skill followed by two performance trials, as recommended in the TGMD-3 and KTK handbooks (*Kiphard & Schilling, 2007*; *Ulrich, 2020*). Children's performances on each skill were videorecorded for assessment.

## Statistical analysis

All the data analysis was conducted using SPSS software (IBM SPSS 26.0; SPSS, Inc., Chicago, IL, USA). Values were considered statistically significant when $p < 0.05$.

Kolmogorov–Smirnova (K–S) was used to test the normality of the outcome parameters. In addition, a histogram, P-P graph, and Q-Q graph were drawn to evaluate the general trend, kurtosis, and skewness values of each resulting parameter for visual inspection. The K–S results showed that the data were not normally distributed ($p > 0.05$), and the histogram, P-P graph, and Q-Q graph also showed that these data had a skewed distribution. Normal transformation attempts to convert non-normal data into normal data, but the data still presented non-normal distribution after transformation. Therefore, the non-parametric test method was adopted in this study.

Spearman's rho correlations were used to examine the test-retest reliability. Cronbach's alpha index was used to verify the internal consistency analysis. Cronbach's alpha values over 0.80 were considered excellent, between 0.70 and 0.80 were considered good, and ranging from 0.60 to 0.69 were considered acceptable (*Cronbach & Meehl, 1955*; *Miller, 1995*).

The criteria validity was examined using concurrent validity, which was computed using Spearman's rho correlation coefficients. Currently, the intensity division of correlation coefficients is not uniform, and the definition of strong, moderate, and low correlation coefficients is different across different research fields and specialities. For example, the definition of a threshold of moderate intensity is 0.4–0.6 in psychology, 0.6–0.7 in

**Table 1 Participants' sociodemographic characteristics, KTK and TGMD-3 scores in the overall sample and sample by gender.**

| Variables | Total | Boys | Girls |
|---|---|---|---|
| *N (%)* | 249 (100.0) | 131 (52.61) | 118 (47.39) |
| *Weight (kg)* | 35.99 (8.60) | 37.80 (9.23) | 33.98 (7.37) |
| *Height (m)* | 1.41 (0.06) | 1.41 (0.06) | 1.41 (0.07) |
| *BMI (kg m$^{-2}$)* | 17.92 (3.43) | 18.82 (3.72) | 16.92 (2.78) |
| KTK | | | |
| *BB* | 31.23 (17.20) | 28.75 (16.42) | 33.99 (17.69) |
| *HH* | 15.99 (15.75) | 17.19 (15.74) | 14.66 (15.72) |
| *JS* | 44.27 (21.18) | 44.63 (21.18) | 43.87 (21.26) |
| *MS* | 18.99 (13.09) | 19.95 (13.99) | 17.92 (11.98) |
| *TS* | 110.49 (46.58) | 110.53 (45.45) | 110.45 (48.00) |
| TGMD-3 | | | |
| *LC* | 36.72 (5.82) | 36.00 (6.48) | 37.52 (4.90) |
| *OB* | 43.65 (5.54) | 45.53 (5.46) | 41.56 (4.85) |
| *TS* | 80.37 (9.21) | 81.53 (10.17) | 79.08 (7.86) |

Note:
Data are expressed as Mean (Standard Deviation); BB, balancing backwards; HH, hopping for height; JS, jumping sideways; MS, moving sideways; TS, Total score; LC, Locomotor skills; OB, Object control skills.

medicine, and 0.3 in politics (*Akoglu, 2018*). In kinesiology, the definition of correlation coefficient strength in the related literature indicated that 0.4–0.5 is a moderate or fair level (*Draghi, Cavalcante Neto & Tudella, 2020*; *Lane & Brown, 2015*; *Menescardi et al., 2022*; *Valentini, Ramalho & Oliveira, 2013*), while a few studies have reported that 0.4–0.5 is a low–level (*Hoeboer, Savelsbergh & De Vries, 2017*). On the whole, statistical values of 0.4–0.5 were acceptable. Due to differences in the intensity division of correlation coefficients, this study did not emphasize the intensity and focused instead on the value of the correlation coefficients.

# RESULTS

## General information

The details of the participants are shown in Table 1. The average BMI of all participants was 17.92, for only boys was 18.82, and only girls was 16.92. For the children's motor competence test, the mean KTK scores of all participants was 110.49, 110.53 for boys, and 110.45 for girls. The mean TGMD-3 score for all participants was 80.37, 81.53 for boys, and 79.08 for girls. Overall, the motor competence level of boys was slightly higher than that of girls.

## Test–retest reliability

The Spearman's rho correlations coefficients of test–retest reliability is shown in Table 2. The overall reliability of the TS, BB, HH, JS, and MS was 0.951, 0.869, 0.918, 0.877, and 0.647, respectively. Specifically, boys reported higher reliability coefficients for the TS (boys = 0.957, girls = 0.934, $p < 0.001$), HH (boys = 0.914, girls = 0.886, $p < 0.001$), and JS

**Table 2 Spearman's rho correlation coefficients of test-retest reliability.**

|  | BB | HH | JS | MS | TS |
|---|---|---|---|---|---|
| Total | 0.869** | 0.918** | 0.877** | 0.647** | 0.951** |
| Boys | 0.837** | 0.914** | 0.875** | 0.660** | 0.957** |
| Girls | 0.850** | 0.886** | 0.871** | 0.666** | 0.934** |

Note:
** Represents $p < 0.001$.
BB, balancing backwards; HH, hopping for height; JS, jumping sideways; MS, moving sideways; TS, Total score.

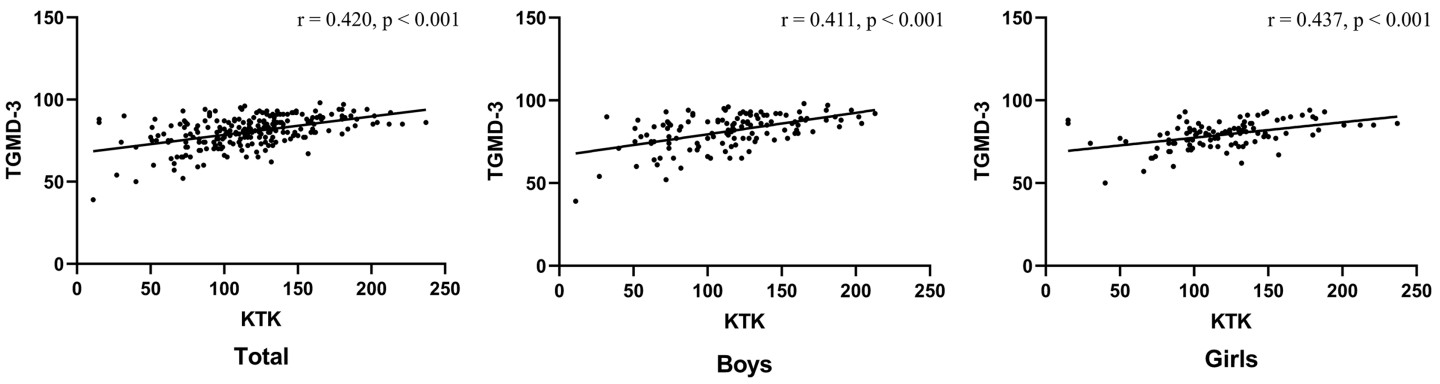

**Figure 2 Spearman's rho correlation coefficient and Scatter plot between the KTK and the TGMD-3.**

(boys = 0.875, girls = 0.871, $p < 0.001$) compared to girls, while girls reported higher coefficients for BB (boys = 0.837, girls = 0.850, $p < 0.001$) and MS (boys = 0.660, girls = 0.666, $p < 0.001$) than boys.

## Internal consistency

The internal consistency was examined by computing Cronbach's Alpha, where 0.60 or higher is an acceptable level (*Cronbach & Meehl, 1955*; *Miller, 1995*).

The overall coefficients were 0.618, 0.583 for boys, and 0.664 for girls. The results confirmed the internal consistency of the KTK, except for the boys. These findings suggested that the KTK is an acceptable instrument to assess motor skills in Chinese children.

## Concurrent validity

The concurrent validity of the KTK was examined by computing Spearman's rho correlations, as shown in Fig. 2. Acceptable validity was found between the overall total (r = 0.420, $p < 0.001$), boys (r = 0.411, $p < 0.001$), and girls (r = 0.437, $p < 0.001$) results from the KTK and TGMD-3.

## DISCUSSION

This was the first study to examine the reliability and validity of the KTK in Chinese children aged 9–10 years. The present study aimed to assess the internal consistency,

concurrent validity, and test–retest reliability of the KTK, and showed that the KTK is a reliable and valid test to assess motor coordination in Chinese children.

In this study, the KTK scores of boys were higher than girls. This finding is consistent with previous research (*Lopes et al., 2011*; *Olesen et al., 2014*; *Re et al., 2018*). Meanwhile, when compared with other countries, Chinese children's KTK scores were slightly lower than those of Australia and Belgium (*Bardid et al., 2015*), but far lower than those in Brazil (*Filho et al., 2021*; *Re et al., 2018*). Overall, the motor coordination level of Chinese children is relatively low.

The results showed a high level of test–retest reliability of the KTK, effectively confirming the test reliability. In this study, the test temporal stability results were high (r = 0.95), which was consistent with the results reported (r = 0.97) in the original edition of the KTK (*Kiphard & Schilling, 2007*). The reliability coefficient of the total score was higher than the score of each test item, which was in line with the findings of previous studies (*Iivonen, Kaarina Sääkslahti & Laukkanen, 2015*). For other studies, the reliability coefficients (r = 0.60–0.99) of the KTK also showed moderate to high-reliability (*Freitas et al., 2015*; *Lopes et al., 2012a, 2011, 2012b*; *Martins et al., 2010*). The KTK showed good reliability in boys and girls, suggesting that it is also suitable for testing in single gender groups, *i.e.*, just boys or girls. Boys outperformed girls in both the HH and JS retests, and girls outperformed boys in both the BB and MS retests.

In this study, the Cronbach's alpha coefficient of KTK in overall children and girls was 0.618 and 0.664, respectively. These results exceed the minimum standard of 0.60 (*Breakweell et al., 2006*). For boys, the Cronbach's alpha coefficient was close to the acceptable level. Therefore, further studies are needed to provide more evidence. The values obtained through the Cronbach's alpha index reflected a good homogeneity profile among the measured subitems and highlighted the internal consistency of the KTK.

For the purpose of obtaining the concurrent validity of the KTK, the TGMD-3 was selected as the reference standard in this study. In terms of reliability and validity, the TGMD-3 is the motor competence assessment tool with the most consistent positive evidence (*Eddy et al., 2020*; *Klingberg et al., 2019*). In addition, the TGMD-3 is the most used tool to test motor competence in the world. It showed good reliability and validity in many countries (*Eddy et al., 2020*; *Griffiths et al., 2018*; *Kim et al., 2014*; *Lopes, Saraiva & Rodrigues, 2016*; *Valentini, 2012*). Concurrent validity showed that a significance and acceptable correlation was found between the KTK and TGMD-3 total scores (r = 0.411–0.437). Compared with other tools, the concurrent validity of this study was relatively lower. For example, the KTK total score showed strong correlations with the Movement Assessment Battery for Children (MABC) total score (r = 0.62–0.65) and the Bruininks–Oseretsky Test of Motor Proficiency Second Edition (BOT-2) total score (r = 0.60–0.64) in Belgian children (*Fransen et al., 2014*; *Henderson & Sugden, 1992*; *Smits-Engelsman, Henderson & Michels, 1998*). The 'fragile' result of this study could be related to the specificity of the TGMD-3 and KTK assessments. The moderate correlation coefficient could be explained by the fact that the TGMD-3 was designed to assess children's gross motor development (*i.e.*, locomotor and object control), while the KTK focused on measures of body and global motor coordination (*Rudd et al., 2016*).

To sum up, the results showed that the KTK is a reliable and valid motor coordination test for Chinese children. It is necessary to make clear the reasons for choosing one assessment tool over another: for example, time, effort, and experience (*Logan et al., 2016*). Process-oriented (*i.e.*, having a combination of process and product) measurement tools tend to take a longer time than product-oriented measuring tools (*Bardid et al., 2019*). The KTK is a typical product-oriented assessment tool. It is considered to have low operational cost, is easy to perform, and is a relatively simple test (*Cools et al., 2009*), which are characteristics that may favour its use for both research purposes and the daily activities of physical education teachers and sports coaches (*Moreira et al., 2019*). At the same time, the KTK is considered to be an effective test battery to assess longitudinal motor competence (*Coppens et al., 2021*), as the motor competence tasks involved in the KTK are characterized by almost no ceiling effect (*Coppens et al., 2021*; *Kiphard & Schilling, 2007*) and are the same for every test item from ages 5 to 15 years old (*Coppens et al., 2021*; *D'Hondt et al., 2013*).

## LIMITATIONS

A limitation of this study was that the analyses of test–retest reliability, internal consistency, and concurrent validity were restricted to children from 9 to 10 years of age. These children were limited in their ability to represent children from 5 to 15 years of age. Therefore, the clinometric results of this version of the test should be interpreted with some caution. Future studies are encouraged to address this limitation and include a full age range. Another limitation is the fact that the participants came from one region-city and one single school, which could have had an impact on the results. Future studies should encourage sampling in multiple regions and schools in China. In addition, the KTK itself also has some shortcomings; for example, only the MS test is a test of object control skills. For the motor competence test, object control is considered to be an important aspect of motor competence in addition to locomotor and balance skills. These three major motor skill areas should be combined in order to evaluate the whole motor competence in a comprehensive method.

## CONCLUSION

This study was the first to demonstrate the validity and reliability of the KTK in Chinese children. The KTK has high test–retest reliability, acceptable internal consistency, and concurrent validity in Chinese children. However, for boys, internal consistency of the KTK should be further examined. In conclusion, the KTK proved to be a valuable instrument for the assessment of the motor coordination of children in China.

### Funding

This work was supported by the Shanghai Science and Technology Planning Project (No. 21010503700) and the Shanghai Key Laboratory of Human Performance (Shanghai

University of Sport, No. 11DZ2261100). The funders had no role in study design, data collection and analysis, decision to publish, or preparation of the manuscript.

## Grant Disclosures

The following grant information was disclosed by the authors:
Shanghai Science and Technology Planning Project: 21010503700.
Shanghai Key Laboratory of Human Performance.
Shanghai University of Sport: 11DZ2261100.

## Competing Interests

The authors declare that they have no competing interests.

## Author Contributions

- Kai Li conceived and designed the experiments, performed the experiments, analyzed the data, prepared figures and/or tables, and approved the final draft.
- Ran Bao conceived and designed the experiments, performed the experiments, analyzed the data, prepared figures and/or tables, and approved the final draft.
- Hyunshik Kim conceived and designed the experiments, authored or reviewed drafts of the article, and approved the final draft.
- Jiameng Ma conceived and designed the experiments, authored or reviewed drafts of the article, and approved the final draft.
- Ci Song performed the experiments, prepared figures and/or tables, and approved the final draft.
- Sitong Chen conceived and designed the experiments, analyzed the data, authored or reviewed drafts of the article, and approved the final draft.
- Yujun Cai conceived and designed the experiments, analyzed the data, authored or reviewed drafts of the article, and approved the final draft.

## Ethics

The following information was supplied relating to ethical approvals (*i.e.*, approving body and any reference numbers):

The Shanghai University of Sport granted Ethical approval to carry out the study (Ethical Application Ref: 102772021RT072).

## Data Availability

The raw measurements are available in the Supplemental Files.

## Supplemental Information

Supplemental information for this article can be found online at http://dx.doi.org/10.7717/peerj.15447#supplemental-information.

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
