# Peer review of "Reliability and validity of the Körperkoordinationstest Für Kinder in Chinese children"

_PeerJ, doi:10.7717/peerj.15447_

## Round 0.1 · original submission · Major Revisions

Dear Authors,

Your work is great and interesting but need some corrections and adjustments prior the publication.

In accordance with the reviewer reports, I suggested modifying the text following their advice which surely iwill mprove the quality of your work.

In particular, English form and grammar need substantial revisions in accordance with the advice provided by the revisors.

Some technical aspects need more clarifications.

This additional comment provided by one reviewer, in my opinion, should not be addressed because the background and the motivation for the KTK choice is well written and clarified by your introduction section.
1) I am not sure how the cultural factors and various contexts of Physical activity referred to Chinese children may reflect on the requested tasks for the KTK test. I don’t understand why you didn’t investigate the reliability, validity and cultural suitability of TGMD3 instead, because it is a measurement tool that can give different results when it is used in different cultural backgrounds (Hulteen et al., 2022).

Reviewer 1 ·

Basic reporting

No comment

Experimental design

No comment

Validity of the findings

The reported findings are relevant to the identified research gap and are supported by the statistical analysis performed by the authors.
Concerning the statistical analysis, the authors should clarify what they mean at line 188-189 by transformation of the normal distribution of the data.
What described in the results and the discussion is coherent with the statistical analysis, and the conclusions are relevant to the study's aim.

·

Basic reporting

Dear Authors, I appreciate your work and contribution to this topic. I will try to help You raise the quality of the work.
English is professional, but I recommend asking a colleague who is an expert in the field and a fluent English speaker to do a language correction because in some places synonyms are used in not clear way.
The introduction clearly shows the context, the gap in knowledge and aims of the study. The literature is adequate and up-to-date.
Please adjust the text and remove single letters like “a” on the end of the lines.

Experimental design

The research meets the aims and scope of the journal. Research questions are well defined. It is stated how research fills an identified knowledge gap.
The research seems to meet technical and ethical standards.

Lines 42-45: there are older and more original (rooted) publications defining physical fitness – it is worth to cite the older as well as the newest approach
Line 134 – please change to “turns” or “tries”
Lines 135-141 – please reframe and add more details – how long was the run up, line 137 – does it mean that in the first attempt there were 5 pillows? And what happened next?
What is more – HH test need coordination to complete it, but it need more strength or dynamic lower limbs force to reach a high result.
Line 145 – please change tests to tries.
What is more – the JS test need coordination but includes also speed and dynamic force of the lower limbs, so it is not fully justyfied to write that the KTK test do not involve physical fitness (line 84)
Lines 155-156 – add what statistical parameters
Line 180 – please change to videorecorded
Lines 210 – 213 – please move to the “participants” section and remove repetitions
Line 268 “The ‘fragile’ result of this study could be related to the specificity of each assessment” – not informative sentence – please reframe
Line 273 – all in all is very colloquial, please change (f.e. To sum up) or remove
Lines 286-294 – please move to a separate section – limitations of the study
Another limitation is the fact that the participants came from one region-city and one single school – this could have had a significant impact on the results.
Lines 296-300 – the conclusions section needs to be totally reframed. Conclusions should be clear and consistent and answer the research questions, confirm the hypothesis or be consistent with the aims of the study. Conclusions should be based only on the results that are presented in the study.
Lines 297-300 can be put in the discussion section

Validity of the findings

Raw data files are provided and can be opened.

Conclusions section needs reframing - see above comments.

There is no information about funding and possible conflicts of interests

Additional comments

The manuscript is well written, it needs minor improvements.

Reviewer 3 ·

Basic reporting

No comment

Experimental design

no comment

Validity of the findings

no comment

Additional comments

Thank you for allowing me to review this manuscript. I believe the authors have carried out an interesting study that could contribute to the literature. I do have some major items that I would like the authors to clarify:

1) I am not sure how the cultural factors and various contexts of Physical activity referred to Chinese children may reflect on the requested tasks for the KTK test. I don’t understand why you didn’t investigate the reliability, validity and cultural suitability of TGMD3 instead, because it is a measurement tool that can give different results when it is used in different cultural backgrounds (Hulteen et al., 2022).

2) I am concerned for the internal validity of your study: in the KTK manual, the authors Kiphard and Schilling, (1974, 2007) wrote that the subtest HH has to be practiced with jump, instead you wrote “run”. These fundamental locomotor skills are really different between them. Can you explore this further? As well as the order that you have done in the execution of the four subtest.


3) The methods section needs a specific re-write/ needs reorganized and corrected.


Here are some specific comments. I hope these comments are helpful to you and assist with the improvement of your manuscript.

Annotated reviews are not available for download in order to protect the identity of reviewers who chose to remain anonymous.

·

Basic reporting

Dear Authors and Editors:
I appreciate your contribution to the preparation of this work, the subject matter discussed in the text is interesting from the point of view of the practical application of an easy and cheap diagnostic method in monitoring children's fitness. It can shed new light and thus increase the potential of the diagnostic tools used (battery tests). However, I have some suggestions and recommendations that can improve the quality of your work and make it easier to interpret.

Basic reporting

The structure and language of the article are good and acceptable, allowing the reader to easily understand its content. However, a few errors in terms of correctness and clarity could be improved by using language correction. This will help to enhance the quality of the language used in the article.
Line 77 - change the wording " a number of"
Line 113 - replace the phase " the aim of our study was"
Line 141-142 - rewrite of clarity
Line 143 - no article "the"
Line 151 - fix the agreement mistake "test" and "points"
Line 160 - change the verb form "were"
Line 162 - change to a genitive case "authors"
Line 168 - no article "the"
Line 181 - change preposition 2x "from"
Line 190 - change the wording " in accordance with"
Line 207 - correct article usage "data" use "the"
Line 218 - change the spelling "specialties" on "specialities"
Line 223 - change the name font "BROWN2" to the same in the references
Line 224 - add a hyphen between "low level"
Line 241 - change the verb form "were" to "was"
Line 271 - add a hyphen between "high reliability"
Line 277,279 - correct article usage "the Cronbach's"
Line 280 - remove the comma after "subitems"
Line 282 - change the wording "In order to"
Line 288 – 289 – Rewrite for clarity
Line 295 – 296 – Rewrite for clarity
Line 305 – change the noun form 2x "assessments"
Line 324 - correct article usage "overall"
Line 324 - change the wording "in a comprehensive manner"

Raw data

I propose making the raw data available in a different file extension, e.g. CSV or Excel because it is a small database that does not contain a large number of rows and columns. Of course, the SAV-type extension is acceptable but is used only in professional statistical analysis programs (commercial). If the work is to be easy to analyze for other scientists, I would suggest changing to CSV - import from this file format is more common in the world.

Experimental design

The main limitation of this validation project is the small number of people tested in relation to the difficulty of performing the test. The test is easy and does not require any financial outlay. Therefore, it was possible to extend the duration of the project and to test more children in different age ranges. The description of the project is easy to repeat for other researchers, so it will certainly be verified in another research group in the future.

Validity of the findings

No major comments. Correct construction of this section.
Only the lack of a section with a clear indication of the further direction of the team's research.

Additional comments

The article is written in accordance with the art and canon of scientific works of international scope. The results allow you to use the knowledge contained in it in two ways. Firstly, it can be easily repeated and criticized by other scientists. Secondly, it provides practical guidance for Physical Education teachers to confirm the value and effectiveness of the test. Nevertheless, it is not a very popular and widely used test in the world to diagnose the fitness level of children and adolescents. Perhaps more time and work with its use is needed to encourage its use.

I Wish You good luck in Your further research!
Michał Nowak, PhD
Jan Długosz University of Humanities and Natural Sciences in Częstochowa
Faculty of health sciences
edukacjawsporcie@icloud.com

---

## Round 0.2 · accepted · Accept

Dear Authors,

Reviewers confirmed that all issues have been addressed and the manuscript is suitable for publication. Congratulations.

Reviewer 1 ·

Basic reporting

No comment

Experimental design

No comment

Validity of the findings

No comment

·

Basic reporting

Dear Authors, Reviewers and Editor
After corrections made in the manuscript by the Authors in response to the reviewers suggestions I find it ready for publication. Thank You all for good cooperation and wish all the best to You and many satisfaction with your further research.

Experimental design

all the suggested corrections have been made

Validity of the findings

all the suggested corrections have been made

Additional comments

all the suggested corrections have been made

Reviewer 3 ·

Basic reporting

No comment

Experimental design

no comment

Validity of the findings

no comment

Additional comments

Reviewer(s)' Comments to Author:

Reviewer: 3 (Anonymus)

Comments to the Author
The authors responded well to my earlier recommendations and criticisms, also adjusting their original manuscript accordingly for which they should be commended.


Recommendation: Accept Submission

First of May, 2023

·

Basic reporting

The corrections have been included

Experimental design

The corrections have been included

Validity of the findings

The corrections have been included

Additional comments

no comments